# DETAILED 3D FACE RECONSTRUCTION IN FULL POSE RANGE

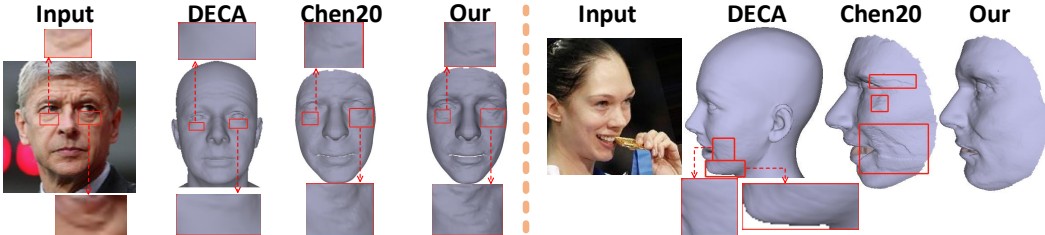

Figure 1: The facial details in small and large poses reconstructed by the existing and our methods.

## ABSTRACT

Monocular detailed 3D face reconstruction aims to recover realistic 3D face from a single-face image. Although existing two-stage reconstruction methods have achieved great success, they are still hard to reconstruct accurate shapes and believable details for large pose images. The reason for the former is that the proportion of large pose data in their training set is often not high, resulting in a limited ability for coarse 3D face reconstruction. The latter is caused by the loss of face details in self-occluded areas of large pose images. In order to perform detailed 3D face reconstruction in full pose range, we respectively propose a self-augment mechanism and a self-supervised detail reconstruction method for large-pose images at the two stages. Specifically, in the first stage, the self-augment mechanism generates a set of large pose data for each training image for re-learning. In the second stage, we pad the self-occluded side of the unwrapped input image according to the face symmetry prior, and design a Recursive Image-to-image Translation Network constrained by the details of input image to estimate its original details. By doing so, we could weaken the training set constraints on coarse 3D face reconstruction and reconstruct the believable face details of large pose images, enabling full pose range detailed 3D face reconstruction. Extensive experiments show that our method could achieve a level comparable to state-of-the-art methods.

## 1 INTRODUCTION

Detailed 3D face reconstruction has made great progress and occupies an important position in the research of computer vision. Generally, the detailed 3D face reconstruction process is divided into two stages: coarse 3D face reconstruction and details reconstruction. Coarse 3D face reconstruction is generally divided into the model-based method and the model-free method, and the model-based method is more widely used, which needs to estimate parameters such as face shape and texture, and then deform the model into an accurate 3D face. Learning these parameter estimations requires a large amount of face data with a large pose to achieve better results. However, the datasets of detailed 3D face reconstruction task have a relatively small proportion of large pose faces which will facilitate the learning of detailed information about the face, but hinders the learning of the relevant parameters representing the 3D structure of the face. For details reconstruction, it is generally divided into methods based on Shape From Shading Horn & Brooks (1989) and methods based on UV displacement map Tran et al. (2018); Chen et al. (2020). The latter has been more widely used in

recent years. The methods based on UV displacement map could be divided into detail reconstruction of small pose and detail reconstruction of large pose according to the application scene. When reconstructing the details of small poses, most of the existing methods could reconstruct the main facial details but still lose some small details, such as small wrinkles which make the reconstruction effect not realistic enough, as shown in the left of figure 1. When reconstructing the details of large poses, most of the existing methods obtain the error facial details, as shown in the right of figure 1. Specifically, when training the reconstruction ability of details, the small pose data could provide complete facial details as labels for self-supervised training, while the occluded areas of large pose data could not provide labels, leading to ineffective training in the ability to reconstruct large pose facial details. Although some existing methods could alleviate these problems, they usually need to introduce additional networks, such as generative adversarial networks, etc., which require a large number of expensive high-resolution training images and are less realistic.

To address these problems, we propose a self-augment mechanism, a Recurrent Image-to-image Translation Network, and a self-supervised large pose details reconstruction method. Specifically, the self-augment mechanism first uses the parameter estimation network to reconstruct the input image into a 3D face, then rotates it to various large poses and renders them as face pictures, and finally returns this group of pictures to the parameter estimation network to reconstruct the 3D face again. Since the large pose data is helpful for the learning of parameter estimation, the parameter estimation ability of the network is self-enhanced, and no new data or network is introduced in the whole process. By doing so, we easily improve the reconstruction capability of the coarse 3D face. It is worth mentioning that since the existing methods use large-scale training sets, even if the proportion of large pose data is not high, its absolute quantity is still not negligible. In order to avoid the possible impact of absolute quantity, we attempt to train our method on small-scale datasets, which makes our conclusions more powerful. For the reconstruction of small pose details, we design a Recurrent Image-to-image Translation Network, which feeds back the output of the decoder to the input of the encoder and aims to strengthen the network's detail reconstruction ability through re-learning. In addition, for the reconstruction of large pose details, our proposed a self-supervised large pose detail reconstruction method, which does not require expensive high-resolution training data. Specifically, we conduct UV unwrapping on the input image and the rendered image of the coarse 3D face, and obtain two unwrapped images that lack the details of self-occluded areas. We pad the content of the missing area and use a Recursive Image-to-image Translation Network to estimate their depth offsets, then get the reconstructed facial details and render it as an image, and finally, we calculate the loss between it and the input image to optimize the estimation of the depth offset. In doing so, the network is able to learn, with label guidance, how to estimate plausible depth offsets for areas of missing detail, thereby reconstructing plausible details for occluded regions of large-pose faces. Thus, we could achieve detailed 3D face reconstruction in full pose range.

## 2 RELATED WORK

### 2.1 COARSE 3D FACE RECONSTRUCTION

In coarse 3D face reconstruction, 3D deformable model (3DMM) based on the principal component analysis proposed by Blanz and Vetter Blanz & Vetter (1999) is a commonly used model. 3DMM is a statistical parametric model including shape and texture parameters, it approximates the 3D face as a linear combination of basic shape and texture. With the development of deep learning, deep neural networks were introduced into 3DMM coefficient estimation Zhu et al. (2016); Tewari et al. (2017); Deng et al. (2019); Feng et al. (2021; 2018a); Jiang et al. (2019); Chen et al. (2020). In these works, 3DDFA Zhu et al. (2016), DAMDNet Jiang et al. (2019), and PRNet Feng et al. (2018a) combined 3DMM, cascaded regression and CNN to align and reconstruct 3D face. Different from them, Chen20 Chen et al. (2020) and DECA Feng et al. (2021) designed a framework for self-supervised 3D face reconstruction. However, most of the existing methods are difficult to guarantee the reconstruction accuracy of large pose face. Specific details are described in the appendix section.

### 2.2 SMALL POSE FACIAL DETAILS RECONSTRUCTION

In recent years, the methods based on UV displacement map have been more widely used. Sela et al. (2017) estimates the high-frequency part of the input image texture to be the UV displacement map to reconstruct the facial details. The facial details reconstructed by Sela et al. (2017) are rough

and accompanied by a large number of noise points. Extreme 3D Tran et al. (2018) uses the details calculated by the SFS method as labels, trains a network to estimate the bump map (displacement map) of the input image and reconstructs it as facial details. Its reconstructions are not real and it could not reconstruct subtle facial details. The UV displacement map estimation of Chen20 Chen et al. (2020) does not require labels and the detail reconstruction effect of Chen20 is more realistic, but some subtle facial details are missed. DECA Feng et al. (2021) uses a generator to generate the UV displacement map for details, which are less realistic. Overall, the common problem of existing methods is that it is difficult to capture subtle facial details. Specific details of the existing methods are in the appendix section.

### 2.3 Large Pose Facial Details Reconstruction

Due to the existence of the self-occlusion phenomenon of large pose faces, the existing solutions usually first complete the missing texture of the self-occlusion area by introducing texture completion modules, and then perform detailed reconstruction like a small pose face. In the existing work of texture completion Zhou et al. (2020); Deng et al. (2018); Gecer et al. (2019); Kim et al. (2021); Chen et al. (2022); Gecer et al. (2021); Wei et al. (2009); Li et al. (2017), Zhou et al. (2020) constructs image pairs for self-supervised learning and training a face texture completion network. von Marcard et al. (2018); Ren et al. (2023) use the GAN Karras et al. (2020) designed for face to perform robust completion for the face area with missing texture. Gecer et al. (2021) generates multi-view face images for a large pose face image, and then stitches them into a complete face texture map. In summary, such solutions need to introduce texture completion modules, such as GAN. However, when training, they require expensive high-resolution training data and face the problem of mode collapse. When inferring, their results are often unstable and unrealistic. Specific details of the existing methods are in the appendix section.

## 3 Method

### 3.1 Overview

As shown in figure 2, our proposed detailed 3D face reconstruction in full pose range is divided into three parts, which are coarse 3D face reconstruction, small pose detailed 3D face reconstruction, and large pose detailed 3D face reconstruction. In the first part, we employ a pre-trained VGG-16 network Simonyan & Zisserman (2014) to estimate the 3DMM coefficients of the input image and reconstruct a coarse 3D face. We then rotate it to multiple large poses and render them as images, feeding them back into the network along with the input image. A set of 3DMM coefficients are estimated and combined to generate a coarse 3D face and render image. In the second part, the render image is fed to the proposed Recursive Image-to-image Translation Network together with the input image to estimate a displacement depth map. We add it to the coarse 3D face obtained in the first stage to generate a detailed 3D face. In the third part, we unwrap the rendered image and the input image, and replace the relatively side part with the relatively frontal part. They are then fed into our Recursive Image-to-image Translation Network to estimate a displacement depth map, resulting in a detailed 3D face.

### 3.2 Coarse 3D Face Reconstruction

**Existing Coarse 3D Face Reconstruction Methods.** The existing coarse 3D face reconstruction methods usually only estimate the 3DMM coefficients once to complete the 3D face reconstruction, so their focus is on how to optimize the coefficient estimation network, which is no problem when the training set contains a high proportion of large pose data. However, the training set of most detailed 3D face reconstruction methods do not contain a high proportion of large pose data, which makes the coarse 3D face not accurate enough.

**Our Coarse 3D Face Reconstruction Method.** In order to improve the coefficient estimation performance of the network under the condition that the training data does not meet the requirements of the network, we propose a self-augment mechanism, as shown in the purple dashed box in the upper left corner of figure 2. 3D face learning in the network needs a large amount of large pose data, the self-augment mechanism uses the identity consistency of the face to perform various large-pose

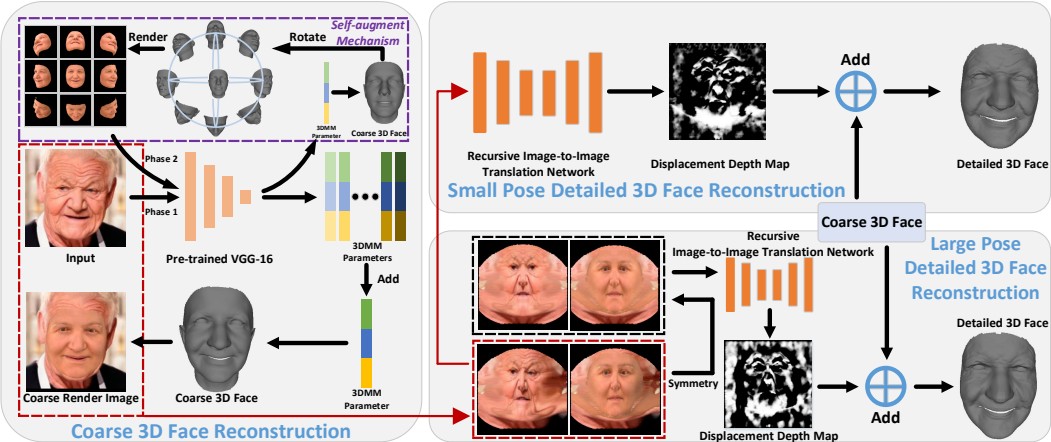

Figure 2: The pipeline of our detailed 3D face reconstruction in full pose range.

rotations on the same face, and then renders them as a large number of large-pose pictures to meet the needs of the network learning. Our network training is divided into two phases. In the first phase, we employ a pre-trained VGG-16 network to estimate the 3DMM coefficients of the input image and obtain a 3D face. The reason for using the pre-trained network is to ensure the basic reliability of obtained 3D face. In the second phase, the self-augment mechanism rotates the 3D face to various large poses and renders the corresponding face images, and then they are fed to the network together with the input image to estimate the 3DMM coefficients of each image. We add these coefficients into one and get a 3D face. We use it to compute the reconstruction loss to constrain the training of the network. In this way, the network's 3DMM coefficient estimation ability could be enhanced, and because this enhancement comes from itself, so our method does not need new training data or networks. In the inference phase, we just load the trained network model without employing the self-augment mechanism. Formally, its workflow is as follows.

$$(\alpha_{id}, \alpha_{exp}, \alpha_{pose}) = VGG(X) \tag{1}$$

$$Pose_i = \{(\delta(\pi/180) \cdot [10 \cdot (6 + i/4)], \delta(160/9) \cdot (6 + i/4)), (0, 0), (0, 0)\} \tag{2}$$

$$V_i = (\alpha_{id} * A_{id} + \alpha_{exp} * A_{exp}) * Pose_i \tag{3}$$

$$(\alpha_{id}, \alpha_{exp}, \alpha_{pose}) = \sum_{n=0}^{1+\max(i)} VGG(Concat(X, Concat_{m=0}^{\max(i)}(Render(V_i)))) \tag{4}$$

$$V = \alpha_{id} * A_{id} + \alpha_{exp} * A_{exp} \tag{5}$$

Where $\alpha_{id}$, $\alpha_{exp}$, $\alpha_{pose}$, $VGG(\cdot)$, $X$, $Pose_i = \{(\cdot), (\cdot), (\cdot)\}$, $V_i$, $A_{id}$, $A_{exp}$, $Render$, $Concat_{m=0}^{\max(i)}(\cdot)$, and $V$ represents 3DMM identity parameter, expression parameter, pose parameter, pre-trained VGG-16 network, input image, rotate parameter with $\{(yaw, tx), (pitch, ty), (roll, sth)\}$, one of the rotated 3D faces, 3DMM identity basis, 3DMM expression basis, render 3D face into image, concatenate $m$ images along batch-size dimension, and final coarse 3D face, respectively. In addition, $\delta = \left\{ \begin{smallmatrix} -1, i \in [0, 12] \\ 1, i \in [13, 25] \end{smallmatrix} \right.$ and $i$ represents the serial number of the pose parameter. Note that the pose parameter could be various, we discuss it in the ablation study section and appendix section.

## 3.3 SMALL POSE DETAILED 3D FACE RECONSTRUCTION

Existing methods usually obtain facial details by a displacement depth map, with the value of each point in the map representing the offset of the corresponding point in the depth direction in the input

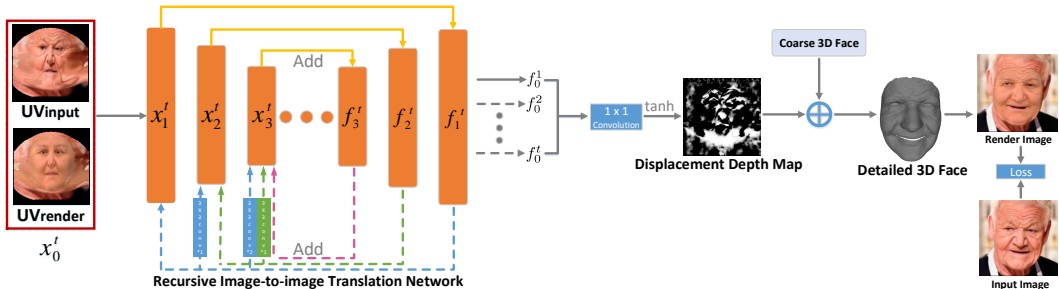

Figure 3: The training pipeline of the proposed RI2ITN. The different colored dotted lines represent the Dense Feedback Connection.

image. However, training the network on a small dataset to obtain this map often leads to network overfitting. In order to alleviate the problem of overfitting, we propose a new skip connection mode to improve the flow of information between encoder layers, that avoids the solidification of feature extraction capabilities by adding beneficial perturbation information to the input of the encoder. Specifically, we introduce feedback connections from any layer to all subsequent layers, as shown in the different colored dotted lines in figure 3. For the output feature maps of the 1st to 9th layers of the decoder, we use a 3×3 convolution kernel to continuously down-sample to multiple scales and then respectively add them and the original feature maps to the corresponding scale feature maps of the encoder. We call this skip connection pattern Dense Feedback Connection ($D^t_{\text{ense}}$), and the network that uses $D^t_{\text{ense}}$ is called Recursive Image-to-image Translation Network (RI2ITN). Formally, the workflow of RI2ITN is as follows.

$$D^t_{ense}(f^t_i) = \begin{cases} f^{t-1}_i + \sum_{j=1}^{i-1} C^{i-j}_{onv}\left(f^{t-1}_j\right) & ,i > 1, t > 1 \\ f^{t-1}_i & ,i = 1, t > 1 \\ 0 & ,i = 0 \\ 0 & ,t = 1 \end{cases} \tag{6}$$

$$x^t_i = E^t_i(x^t_{i-1}) + D^t_{ense}(f^{t-1}_{i-1}) \tag{7}$$

$$f^t_i = D^t_i(f^t_{i+1}) + x^t_i \tag{8}$$

$$f^t_0 = D_{econv}(f^t_1) \tag{9}$$

$$M_{disp} = tanh(C_{onv1\times1}(C_{oncat}(f^1_0, ..., f^t_0))) \tag{10}$$

where $x^t_0$, $f^t_0$, and $D_{econv}(\cdot)$ respectively denote the input of the encoder, the output of the decoder, and $3 \times 3$ transposed convolution that the number, stride, and padding of convolution kernels is 1, 1, and same, respectively. Moreover, $C_{onv1\times1}(\cdot)$, $C_{oncat}(\cdot)$, $M_{disp}$ represent $1 \times 1$ convolution that the number of convolution kernels is one, concatenate operation, and the displacement depth map, respectively. In this way, our RI2ITN finely re-optimizes the extracted information to learn the discriminative facial features and produce a finer displacement depth map. In addition, $C^{i-j}_{onv}(\cdot)$ represents the down-sampling operations of $i - j$ consecutive $3 \times 3$ convolution with Relu activation function. By doing so, our Dense Feedback Connections continuously perturb the encoder to avoid the immobilization of feature extraction capabilities, thereby alleviating the overfitting problem caused by the small-scale training set.

## 3.4 LARGE POSE DETAILED 3D FACE RECONSTRUCTION

Since most of the existing methods require additional texture completion modules which are costly, the large pose detailed reconstruction has not been properly resolved. If following the idea of working on the texture of most existing methods, it will bring a huge resource overhead, so we propose

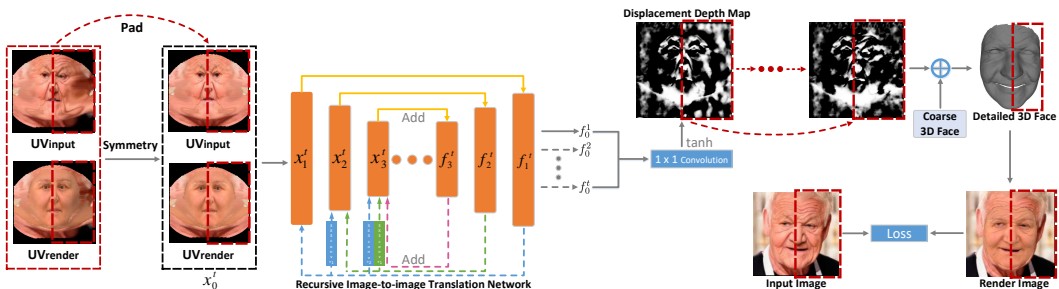

Figure 4: The training pipeline of our self-supervised detail reconstruction method for images with large poses.

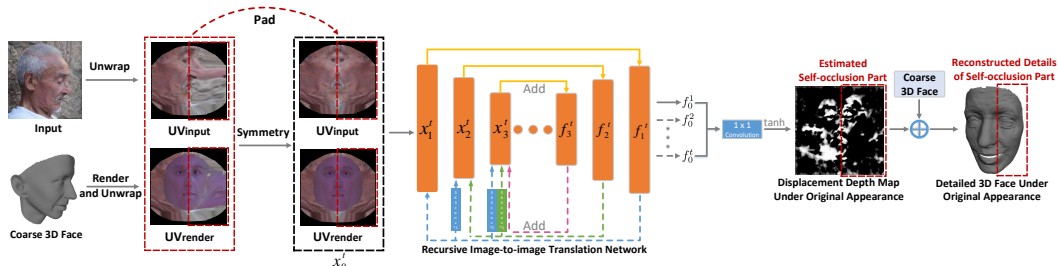

Figure 5: The infer pipeline of our self-supervised detail reconstruction method for images with large poses.

the idea of working on the displacement depth map, without the texture completion steps. As shown in the red dotted line box in figure 4, the left side of the two unwrapped images is the part that is biased toward the front face, and we use them to symmetrically replace the right part. The reason for symmetry is that the geometric structure of the face has a symmetrical nature, and learning based on symmetrical faces could obtain better guidance. Then the two unwrapped images are input to our RI2ITN to estimate the displacement depth map, and then reconstruct the detailed 3D face. We calculate the loss between the image rendered by the detailed 3D face and the input image to constrain the learning of the network. In the replaced position, the facial details reconstructed by the network in the early stage do not conform to the input image. Gradually, the network could adaptively learn the displacement depth map before being replaced, so as to reconstruct the facial details that conform to the input image. Our method uses those small pose data of the small-scale dataset. Because the datasets used by the detailed 3D face reconstruction task contain a large proportion of small pose data, which could maintain complete texture information after UV unwrapping, and our method could use them as reconstruction labels to enable the method to work, so our method could be effectively trained. When the input image is a large pose face, the self-occlusion part will have no label, but this will not cause harm, Because the reconstructed 3D face will be rendered to 2D image with the large pose of input image, this image no longer contains the occluded facial parts. Therefore, when calculating the loss between this image and the input image, the occluded facial parts are not involved in the loss computation. Through this self-supervised training method, the network could estimate the displacement depth map of the original face for the replaced face, and thus obtain the original facial details. As shown in figure 5, during inference, we unwrap the large pose input image and the rendered image of coarse 3D face, and replace symmetrically the relatively side part with the relatively frontal part. After receiving the two unwrapped images, the network will estimate the displacement depth map of the original appearance, and finally, obtain the believable facial details of the self-occlusion part.

## 4 OBJECTIVE FUNCTION

**Coarse 3D Face Reconstruction.** The total objective function is denoted as:

$$\mathcal{L} = w_1 \mathcal{L}_p + w_2 \mathcal{L}_{lm} + w_3 \mathcal{L}_{id} + w_4 \mathcal{R}_{param} \tag{11}$$

where $\mathcal{L}_p$ is the pixel loss and it calculates the average $L_{2,1}$-distances between the value of each pixel of the input image and 2D image which is rendered by the coarse 3D face. $\mathcal{L}_{lm}$ is the landmark consistency loss, which measures the average $L_2$-distance between the GT 68 2D landmarks of input image and the 68 landmarks of the reconstructed 3D face. $\mathcal{L}_{id}$ is the perceptual identity loss and reflects the perception similarity between the two images. We send both the input image and the image rendered by coarse 3D face to VGG-16 to extract features, and then calculate the $L_2$-distance between the two extracted feature vectors. In addition, $\mathcal{R}_{param}$ represents the regularization term for 3DMM parameters estimated by VGG-16. The weights $w_1$, $w_2$, $w_3$, and $w_4$ are constant values to balance the influence of each loss term. Specific objective function details are in the appendix section.

**Detailed 3D Face Reconstruction.** We adopt the same objective functions to train RI2ITN in both small and large pose reconstructions. The total objective function is denoted as:

$$\mathcal{L} = a_1 \mathcal{L}_p + a_2 \mathcal{L}_s + a_3 \mathcal{R}_{disp} \tag{12}$$

where $\mathcal{L}_p$ is the pixel loss and it calculates the difference in the value of each pixel of the input image and 2D image which is rendered by the detailed 3D face. $\mathcal{L}_s$ is the smoothness loss that is employed on both the UV displacement normal map and facial displacement depth map. In addition, $\mathcal{R}_{disp}$ is a regularization term for smoothness loss to reduce severely depth changes, which may introduce distortion in the face on the 3D mesh.

$$\mathcal{L}_s = \sum_{i \in \mathcal{V}_{UV}} \sum_{j \in \mathcal{N}(i)} b_1 \|\Delta n(i) - \Delta n(j)\|^2 \\ + b_2 \|\Delta z(i) - \Delta z(j)\|^2 \tag{13}$$

where $\mathcal{V}_{UV}$ are vertices in the UV space and $\mathcal{N}(i)$ is the neighborhood of vertex i with a radius of 1. $\Delta n(i)$ and $\Delta n(j)$ represent the difference of UV normal map before and after adding displacement depth map for $i$ and $j$, respectively. $\Delta z(i)$ and $\Delta z(j)$ represent the difference in displacement depth map before and after adding UV normal map for $i$ and $j$, respectively. Therefore, the smoothness loss $\mathcal{L}_s$ ensures a similar representation of the neighboring pixels on these maps and the robustness to mild occlusions. The weights $a_1$, $a_2$, $a_3$, $b_1$, and $b_2$ are constant values to balance the influence of each loss term.

## 5 EXPERIMENT

### 5.1 IMPLEMENTATION DETAILS

We describe the specific implementation details in the appendix section.

### 5.2 DATASETS

**Training Datasets.** Our training dataset consists of 1,000 images from the training set of the CelebA dataset Liu et al. (2016). Specifically, our self-augment mechanism addresses the issue of poor performance in large pose scenarios by augmenting the limited number of large pose images. Although the proportion of with large pose images in the training dataset is relatively low, their absolute count is still not insignificant. Considering that the number of large pose images in the training set may affect the effectiveness verification of our self-augment mechanism, we select a small-scale dataset for training. In this setup, we train the detailed 3D face reconstruction methods Chen20 Chen et al. (2020) and DECA Feng et al. (2021) using 1,000 images for comparison in experiment section to ensure fairness.

**Evaluation Datasets.** We use MICC Florence Bagdanov et al. (2011), NOW Validation Sanyal et al. (2019), 3dr Booth et al. (2018), and 3ds Booth et al. (2018) for evaluation. Specific datasets details and division details are in the appendix.

### 5.3 QUANTITATIVE EVALUATION

To verify the effectiveness of the methods in small and large postures, we divide each dataset into two indicators of small pose and large pose for evaluation. We use the evaluation protocol proposed by Feng et al. (2018b) to calculate the Normalized Mean Error (NME) for evaluation. Table 1 shows the quantitative results of our method and existing coarse 3D face reconstruction methods.

Table 1: Coarse 3D face reconstruction quantitative results in two poses on evaluation datasets.

| Method / NME | MICC Small Pose | MICC Large Pose | NOW Small Pose | NOW Large Pose | 3dr Large Pose | 3ds Large Pose | Mean |
|---|---|---|---|---|---|---|---|
| DAMDNet | 3.345±2.824 | 3.538±2.837 | 3.357±2.829 | 3.784±3.100 | 3.252±2.108 | 3.733±2.392 | 3.501±2.681 |
| Extreme 3D | 3.219±3.261 | 3.407±3.275 | 3.437±3.361 | 3.685±3.406 | 3.547±2.991 | 3.208±2.252 | 3.417±3.091 |
| 3DDFA | 3.314±2.762 | 3.262±2.728 | 3.305±2.811 | 3.405±2.765 | 3.055±2.120 | 3.254±2.120 | 3.265±2.551 |
| Chen20 | 2.510±2.176 | 3.177±2.618 | 2.824±2.468 | 3.559±2.928 | 2.413±1.915 | 2.459±1.778 | 2.823±2.313 |
| PRNet | 2.943±2.139 | 2.604±2.003 | 3.281±3.026 | 3.075±2.835 | 2.437±2.249 | 2.547±2.059 | 2.814±2.385 |
| DECA | 2.478±2.374 | 2.587±2.425 | 2.402±2.334 | 2.628±2.507 | 2.325±2.184 | 2.608±2.309 | 2.504±2.355 |
| **Our** | **2.352±2.010** | **2.476±2.109** | **2.317±2.019** | **2.341±1.992** | **2.225±1.974** | **2.178±1.779** | **2.314±1.980** |

Compared with DECA Feng et al. (2021) and PRNet Feng et al. (2018a), which are the 2D and 3D supervised state-of-the-art methods, respectively, we achieved surpassing in all metrics, proving our method outperforms them. Furthermore, the existing methods exhibit significant error in all large pose indicators, strongly substantiating our claim that the existing methods encounter difficulties in accurately reconstructing 3D shapes from images with large pose variations. In contrast, our method achieves the lowest errors across all large pose indicators, demonstrating its efficacy in addressing large pose scenarios.

Table 2: Detailed 3D face reconstruction quantitative results for same face topological model.

| Method / P2P Error | MICC Small Pose | MICC Large Pose |
|---|---|---|
| Extreme 3D | 1.985±1.523 | 2.028±1.596 |
| Chen20 | 1.802±1.369 | 2.123±1.597 |
| **Our** | **1.521±1.105** | **1.570±1.148** |

Table 3: Detailed 3D face reconstruction quantitative results for different face topological model.

| Method / NME Error | MICC Small Pose | MICC Large Pose |
|---|---|---|
| DECA | 2.694±2.545 | 2.765±2.563 |
| **Our** | **2.543±2.392** | **2.653±2.474** |

In terms of detailed 3D face reconstruction quantitative, following Chen20 Feng et al. (2021), we compute the point-to-point (p2p) distance error on the MICC dataset to compare methods utilizing the same facial topology model, as shown in Table 2. To compare methods utilizing the different facial topology model, following Ren et al. (2023) and DECA Feng et al. (2021), we perform rigid alignment and cropping of the ground truth scan and the predicted 3D faces. Subsequently, we measure the distance between all ground truth scan vertices to the closest points on the predicted 3D faces surface, as shown in Table 3. From these two tables, it is evident that our method demonstrates the lowest errors. This emphasizes that our small and large pose detailed 3D face reconstruction methods could capture and reconstruct more accurate facial details.

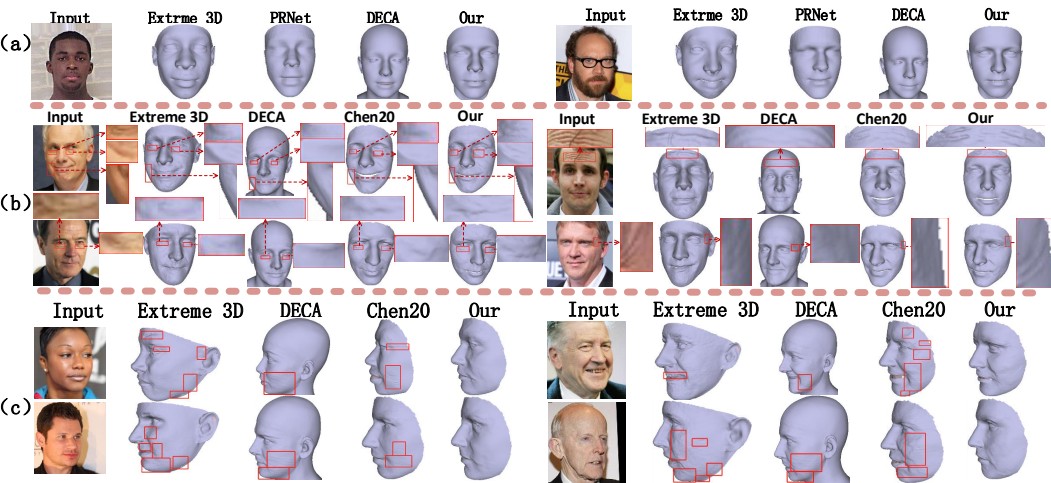

Figure 6: The qualitative evaluation results, where (a), (b), and (c) represents the coarse, small pose detailed, and large pose detailed 3D face reconstruction results, respectively.

## 5.4 QUALITATIVE EVALUATION

We show the qualitative experimental results of coarse, small pose detailed, and large pose detailed 3D face reconstruction in the (a), (b), and (c) of figure 6, respectively. As shown in the second example in figure 6(a), the eyes reconstructed by Extreme 3D and PRNet do not match the original image, and DECA incorrectly reconstructs the mouth corner as an upward curl. In contrast, our method accurately reconstructs the shapes of the eyes and mouth corners. This shows that our method has achieved effective improvements in both shape and expression reconstruction. As illustrated in the second column, second row of figure 6(b), the three existing methods have not captured the crow's feet of the input at all, whereas our method faithfully reproduces the shape of these crow's feet. This shows that our method has a stronger details reconstruction ability of small pose face. For the reconstruction of face details in self-occlusion areas in large poses, as shown in the first row, second column of figure 6(c), Extreme 3D reconstructs the teeth to the lips, DECA has stripes, and the Chen20 has many messy details, while our result is the most believable and has rich details. This show that our method could reconstruct more believable details for large pose faces. Combining (a), (b), and (c) in figure 6, it could be seen that our method achieves reliable and detailed 3D face reconstruction in full pose. Furthermore, to verify the generalization of our method, we conduct qualitative experiments on other two datasets Yang et al. (2020) and Yin et al. (2006), as shown in the appendix section.

## 5.5 ABLATION STUDY

Table 4: Ablation study of coarse 3D face reconstruction.

| Method / NME | MICC Small Pose | MICC Large Pose | NOW Small Pose | NOW Large Pose | 3dr Large Pose | 3ds Large Pose | ESRC Small Pose | ESRC Large Pose |
|---|---|---|---|---|---|---|---|---|
| Our w/o SAM | 2.510±2.176 | 3.177±2.618 | 2.824±2.468 | 3.559±2.928 | 2.413±1.915 | 2.459±1.778 | 2.780±2.492 | 3.570±3.227 |
| Our | 2.446±2.050 | 2.746±2.266 | 2.430±2.210 | 2.765±2.325 | **2.224±1.907** | **2.127±1.622** | 2.217±2.036 | 2.974±2.714 |
| **Our∗** | **2.352±2.010** | **2.476±2.109** | **2.317±2.019** | **2.341±1.992** | 2.225±1.974 | 2.178±1.779 | **2.136±1.922** | **2.378±2.159** |

We show the ablation study of coarse 3D face reconstruction in Table 4. Where SAM represents the self-augment mechanism, while Our and Our∗ refer to our method respectively using two different sets of SAM parameters. self-augment mechanism greatly expands the large pose training data, which meets the needs of the network for large pose data, thus effectively improving the network's 3DMM coefficient estimation ability. In addition, different parameters lead to different improvements, the parameters details used by Our and Our∗ are described in the implementation details section of Appendix.

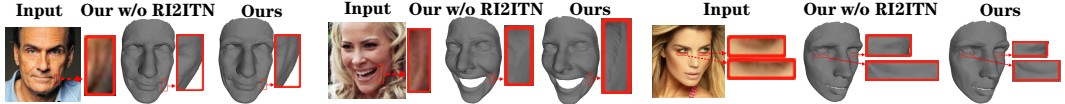

Figure 7: The qualitative ablation results of our detailed 3D face reconstruction method.

Due to the fact that facial details contribute only a small proportion to the quantitative error, we conduct qualitative ablation experiments to more clearly demonstrate the effectiveness of our detailed 3D face reconstruction method, as shown in figure 7. It is evident that with the introduction of RI2ITN, our method is capable of observing and capturing numerous subtle details. This validates the effectiveness of our detailed 3D face reconstruction method.

## 6 CONCLUSION

Aiming at the issues of detailed 3D face full pose reconstruction in existing methods, this paper proposes a self-augment mechanism and a self-supervised detail reconstruction method for large pose images. Our method effectively improves the reconstruction accuracy of coarse 3D faces without introducing new training data and networks. For the reconstruction of the facial details in large pose , we convert the texture completion into the estimation of the displacement depth map, which greatly reduces the complexity and improves the reconstruction effect. Extensive experiments demonstrate the effectiveness of our method.

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

## A    RELATED WORK

### A.1    COARSE 3D FACE RECONSTRUCTION

For coarse 3D face reconstruction, the general route is divided into model-based methods and model-free methods. Among them, the application of model-based methods is more extensive, the 3D deformable model (3DMM) based on the principal component analysis proposed by Blanz and Vetter Blanz & Vetter (1999) is a commonly used model. In work based on the BFM models, 3DDFA combined 3DMM, cascaded regression and CNN to align and reconstruct 3D face. In this process, alignment and reconstruction are done together, and 3DDFA fits the entire 3D dense face model to the face picture instead of the sparse feature point shape. Different from 3DDFA, MOFA proposes a framework for self-supervised 3D face reconstruction. Its input is a picture and 2D key points (optional), and its encoder extracts the pose, shape, expression, facial reflection, and illumination of the picture, and uses these parameters to reconstruct a 3D face. During training, the reconstructed 3D face is projected back to 2D, and a loss is computed with the input image to optimize the reconstruction process. Since most 3D face reconstruction methods use synthetic data or 3D labels, the

quality of the data will affect the accuracy of the reconstruction. To alleviate this problem, Deng et al. (2019) proposes a hybrid loss function for weak supervision, which improves the reconstruction accuracy by integrating pixel loss and perceptual loss. In work based on the FLAME model, DECA decouples camera, albedo, light, shape, pose, and expression parameters for the input image, and loads them into the FLAME model to obtain a 3D face. Similar to MOFA, DECA finally projects the 3D face into a 2D image for self-supervised training, and DECA also has 2D keypoint labels for weakly supervised training. In order to obtain more realistic 3D face, MICA employs 3D labels for supervised training. However, these methods need expensive large-scale training set to achieve good results, otherwise, their reconstruction accuracy will be difficult to guarantee.

## A.2    SMALL POSE FACIAL DETAILS RECONSTRUCTION

For the detailed reconstruction of small pose face, it is generally divided into methods based on Shape From Shading and methods based on UV displacement map. The latter has been more widely used in recent years. Sela17 estimates the high-frequency part of the texture by subtracting the low-pass filtered part from the luminance values of the input image, which could be obtained by convolving the texture with a spatially varying Gaussian kernel in the image domain. Then Sela17 uses the high-frequency part of the texture to reconstruct the details of the face, but in practical applications, the details of the face reconstructed by Sela17 are rough and accompanied by a large number of noise points. Extreme 3D uses the details calculated by the SFS method as labels, trains a network to estimate the bump map (displacement map) of the input image and reconstructs it as facial details. The disadvantage of Extreme 3D is that the facial details it reconstructs are not real and it could not reconstruct small facial details. The UV displacement map of chen20 is obtained by the difference between the input image and the coarse face rendering image, and the input image is used as the self-supervised training label. The detail reconstruction effect of chen20 is more realistic, but some subtle facial details are missed. DECA does not estimate the UV displacement map from the input image, but uses a generator to generate it. DECA could not realistically restore the facial details of the input image. Unlike methods using UV displacement maps, based on self-supervised decomposition of diffuse normals and specular normals, Ren et al. (2023) uses SFS to obtain detailed facial geometry and approximate diffuse albedo, diffuse shadows, and specular shadows. However, the details reconstructed by Ren et al. (2023) are sparse, and there are obvious noise points. Overall, the common problem of existing methods is that it is difficult to capture subtle details in human faces.

## A.3    LARGE POSE FACIAL DETAILS RECONSTRUCTION

Due to the existence of the self-occlusion phenomenon of large pose faces, the existing solutions usually first complete the missing texture of the self-occlusion area by introducing texture completion modules, and then perform detailed reconstruction like a small pose face. In the existing work of texture completion Zhou et al. (2020); Deng et al. (2018); Gecer et al. (2019); Kim et al. (2021); Chen et al. (2022); Gecer et al. (2021); Wei et al. (2009); Li et al. (2017), Wei et al. (2009) apply a Markov random field model to infer missing local textures from global textures. With the boom in deep learning, neural networks were introduced to this task. The face completion network of Li et al. (2017) is trained through a local discriminator, a global discriminator, and a fixed analysis network. Zhou et al. (2020) constructs image pairs for self-supervised learning and training a face texture completion network. Deng et al. (2018) replaces the parsing network of Li et al. (2017) with a fixed identity classification network and trains a network to fill in missing textures. von Marcard et al. (2018); Ren et al. (2023) use the GAN Karras et al. (2020) designed for face to perform robust completion for the face area with missing texture. Gecer et al. (2021) generates multi-view face images for a large pose face image, and then stitches them into a complete face texture map. In summary, such solutions need to introduce texture completion modules, such as GAN. However, when training, they require expensive high-resolution training data and face the problem of mode collapse. When inferring, their results are often unstable and unrealistic.

## B    OBJECTIVE FUNCTION

### B.0.1    COARSE 3D FACE RECONSTRUCTION

The total objective function is denoted as:

$$\mathcal{L} = w_1\mathcal{L}_p + w_2\mathcal{L}_{lm} + w_3\mathcal{L}_{id} + w_4\mathcal{R}_{param} \tag{14}$$

Specifically, the every objective function is denoted as:

$$\mathcal{L}_p = \frac{1}{|\mathcal{M}|}\sum_{(i,j)\in\mathcal{M}}\left\|I_{i,j}-I_{i,j}^R\right\|_2, \tag{15}$$

where $M$ are the pixels of the visible region on the $I^R$, while $i$ and $j$ are their positions. $\mathcal{L}_p$ is obtained by averaging the $L_{2,1}$ distance of all visible pixels.

$$\mathcal{L}_{lm} = \frac{1}{N}\sum_{i=1}^{N}\left\|p_i-p_i^R\right\|_2^2 \tag{16}$$

where $p_i$ represents the $i$-th landmark position in the input image, $p_i^R$ is the corresponding $i$-th landmark position in the rendered face image, and N = 68 is the number of landmarks. $\mathcal{L}_{lm}$ could constrain the network's learning of pose and expression parameters.

$$\mathcal{L}_{id} = \left\|\phi(I)-\phi\left(I^R\right)\right\|_2^2 \tag{17}$$

where $\phi$ represents the feature extraction process of the VGG-16 network.

$$\mathcal{R}_{param} = \omega_s\left\|x_{shape}\right\|^2 + \omega_e\left\|x_{exp}\right\|^2 \tag{18}$$

where $\omega_s$ and $\omega_e$ are weighting parameters.

### B.0.2    DETAILED 3D FACE RECONSTRUCTION

The total objective function is denoted as:

$$\mathcal{L} = a_1\mathcal{L}_p + a_2\mathcal{L}_s + a_3\mathcal{R}_{disp} \tag{19}$$

Specifically, $\mathcal{L}_p$ is the same as the $\mathcal{L}_p$ in Coarse 3D Face Reconstruction section, $\mathcal{L}_s$ is introduced in the body of the paper, and $\mathcal{R}_{disp}$ is denoted as:

$$\mathcal{L}_{disp} = \sum_i w_{dn}\|\Delta n(i)\|^2 + w_{dz}\|\Delta z(i)\|^2 \tag{20}$$

This regularization term could alleviate the drastic depth value variation in the estimation of the displacement depth map, thereby reducing the distortion of the reconstructed 3D face.

## C    IMPLEMENTATION DETAILS

The number of training steps for the coarse, small pose detailed, and large pose detailed 3D face reconstruction is 250, 20000, and 5000, respectively. We first train the Chen et al. (2020) method for 20,0000 steps on the small training set, and then employ its VGG-16 as the network of our coarse 3D face reconstruction. The number of training steps for the coarse 3D face reconstruction is 250 steps, and the batch size is 6.

In the self-augment mechanism, we set two different parameter settings, as shown in Table 4 of ablation study. In the first parameter setting we use for Our$*$ in Table 4, the rendering number of large pose faces is set to 26, while the pose parameter yaw $yaw_i$ and its displacement $tx_i$ are set to $\left\{\begin{smallmatrix}yaw_i=\delta(\pi/180)\cdot[10\cdot(6+i/4)]\\tx_i=\delta(160/9)\cdot(6+i/4)\end{smallmatrix}\right.$ , pitch $pitch_i$ and its displacement $ty_i$ are set to $\left\{\begin{smallmatrix}pitch_i=0\\ty_i=0\end{smallmatrix}\right.$ , roll $roll_i$ and its displacement $sth_i$ are set to $\left\{\begin{smallmatrix}roll_i=0\\sth_i=0\end{smallmatrix}\right.$ , where $\delta=\left\{\begin{smallmatrix}-1,i\in[0,12]\\1,i\in[13,25]\end{smallmatrix}\right.$ and $i$ represents the serial number

of the pose parameter. In the second parameter setting we use for Our in Table 4, the rendering number of large pose faces is set to 9, while the pose parameter yaw $yaw_i$ and its displacement $tx_i$ are set to $\begin{cases} yaw_i = \delta(\pi/180) \cdot (-45), i \in [2,4] \\ tx_i = \delta(-80), i \in [2,4] \end{cases}$ and $\begin{cases} yaw_i = 0, i \in \{0,1,5\} \\ tx_i = 0, i \in \{0,1,5\} \end{cases}$ , pitch $pitch_i$ and its displacement $ty_i$ are set to $\begin{cases} pitch_i = \delta(\pi/180) \cdot (45), i \{1,2,8\} \\ ty_i = \delta(-80), i \{1,2,8\} \end{cases}$ and $\begin{cases} pitch_i = 0, i \{0,3,7\} \\ ty_i = 0, i \{0,3,7\} \end{cases}$ , roll $roll_i$ and its displacement $sth_i$ are set to $\begin{cases} roll_i = 0 \\ sth_i = 0 \end{cases}$ , where $\delta = \begin{cases} -1, i \{4,5,6\} \\ 1, i \{1,2,8\} \end{cases}$ and $i$ represents the serial number of the pose parameter.

In the training of the small pose detailed 3D face reconstruction, due to the fact that increasing the value of "t" of RI2ITN beyond 2 hardly yields any additional benefits in terms of detail reconstruction, it instead results in increased resource consumption. Therefore, we have set $t$ of RI2ITN to 2, the number of training steps is 20,000. In the training of the large pose detailed 3D face reconstruction, we load the RI2ITN trained by the small pose detailed 3D face reconstruction, and continue to train for 5000 steps. We write the codes with TensorFlow 1.15 and train on a Tesla V100 and a GTX 2080Ti for coarse and detailed face reconstruction, respectively. The optimizer and all hyperparameter settings such as the objective function weights $w_1, w_2, w_3, w_4, \omega_s, \omega_e, a_1, a_2, a_3, b_1$, and $b_2$ are all consistent with the open source version of the Chen et al. (2020). The code of Chen et al. (2020) is available from `https://github.com/cyj907/unsupervised-detail-layer`. The Chen et al. (2020) mentioned in the experimental part of this paper was trained on the small-scale dataset using its open source version. The epoch of DECA Feng et al. (2021) trained through small-scale dataset is set to 10 times to published version, and other all hyperparameter settings are the same to the published version. The code of DECA Feng et al. (2021) is available from https://deca.is.tue.mpg.de/.

## D  DATASETS

### D.0.1  TRAINING DATASETS

CelebFaces Attributes Dataset (CelebA) is a large-scale face attributes dataset with more than 200K celebrity images. We use 1,000 images from the training set of the CelebA dataset.

### D.0.2  EVALUATION DATASETS

- MICC Florence. In MICC, videos are taken on 53 subjects under Indoor Cooperative, PTZ Indoor, and PTZ Outdoor conditions. The ground truth 3D scans are provided for 52 out of the 53 people. We select the Indoor Cooperative condition which has the clearest face, and crop a face picture with a small pose and a face picture with a large pose for each available subject from the clear video frame as the evaluation dataset.

- NOW Validation. In this dataset, there are 20 subjects in total, each subject has face pictures from 5 or 6 poses and a ground truth 3D scan. For each subject, we select a face picture with small pose and a face picture with large pose as the evaluation dataset under the two poses respectively.

- 3dr. It is an abbreviation for 3dMDLab-real dataset and is a sub-sets of 3dMDLab dataset Booth et al. (2018). It includes 8 real images in large pose, coming directly from one of the RGB cameras of the 3dMD face scanning system. The images are high-resolution (2048x2448 pixels) images with true colour range (24 bits per pixel).

- 3ds. It is an abbreviation for 3dMDLab-synthetic dataset and is a sub-sets of 3dMDLab dataset Booth et al. (2018). It includes 6 synthetic images in large pose created by the same scans after rendering them from different view points with varying synthetic light. Again, the images are high-resolution (2048x2448 pixels) images with true colour range (24 bits per pixel).

- ESRC. It Feng et al. (2018b) contains about 6 pose images of 135 subjects, and a reference 3D face scan for each subject. we select a face picture with small pose and a face picture with large pose as the evaluation dataset under the two poses respectively.

# E    EVALUATION BENCHMARK

For coarse 3D face reconstruction, we use the evaluation protocol proposed by Feng et al. (2018b) to calculate the Normalized Mean Error (NME), which refers to the normalized mean of the distances between all reference scan vertices to the closest points on the reconstructed mesh surface, after rigidly aligning scan and reconstruction. For detailed 3D face reconstruction, following Chen et al. (2020), in order to better reconstruct facial details, we use 52 clear face images with small pose cropped from MICC as input images. We cut the reconstructed 3D face and ground truth 3D scan to 95mm centered on the tip of the nose, and then perform rough rigid alignment on the two based on 7 key points, and then run iterative closest point algorithm to perform precise rigid alignment, and finally calculate the point-to-point distances error of the two.

# F    EXPERIMENT

## F.1    QUALITATIVE EVALUATION

We show more results of details reconstruction in figure 8. It could be seen that our method captures and reconstructs finer wrinkles, which are often lost by other methods.

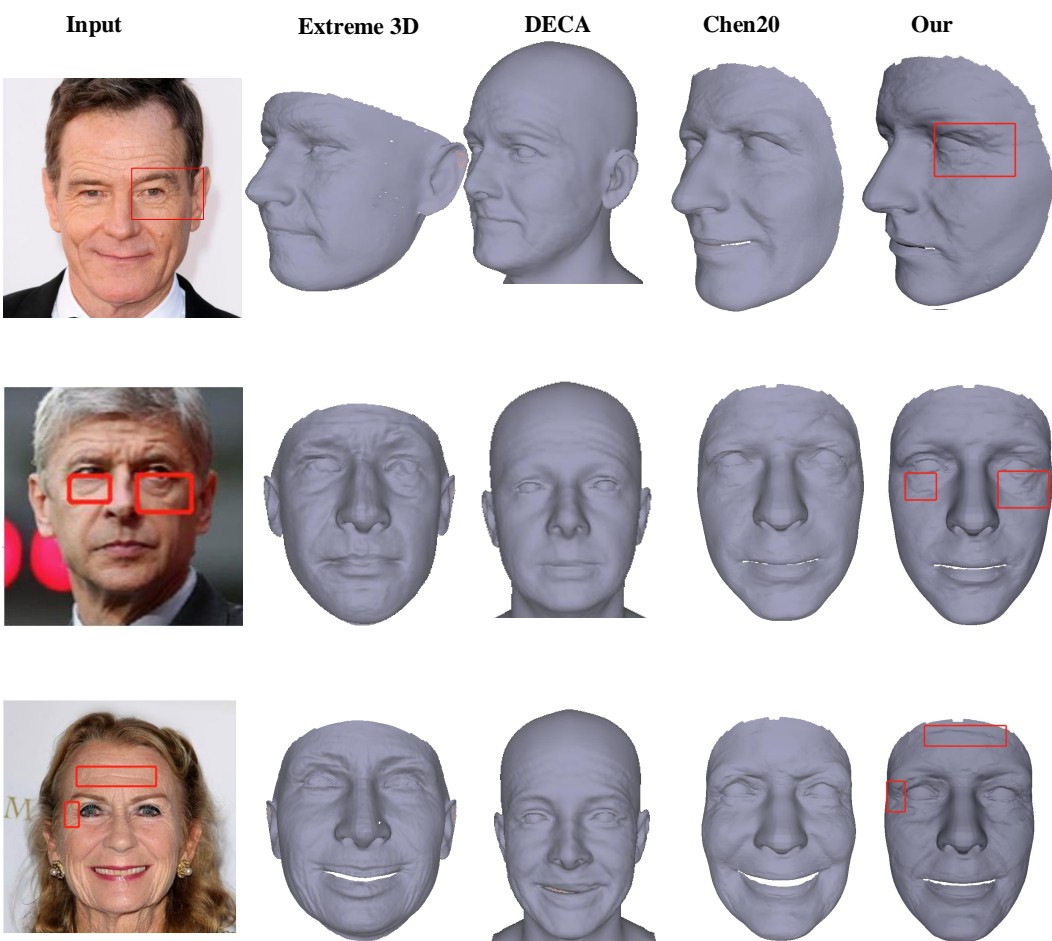

Figure 8: Detailed 3D face reconstruction results of the existing and our methods.

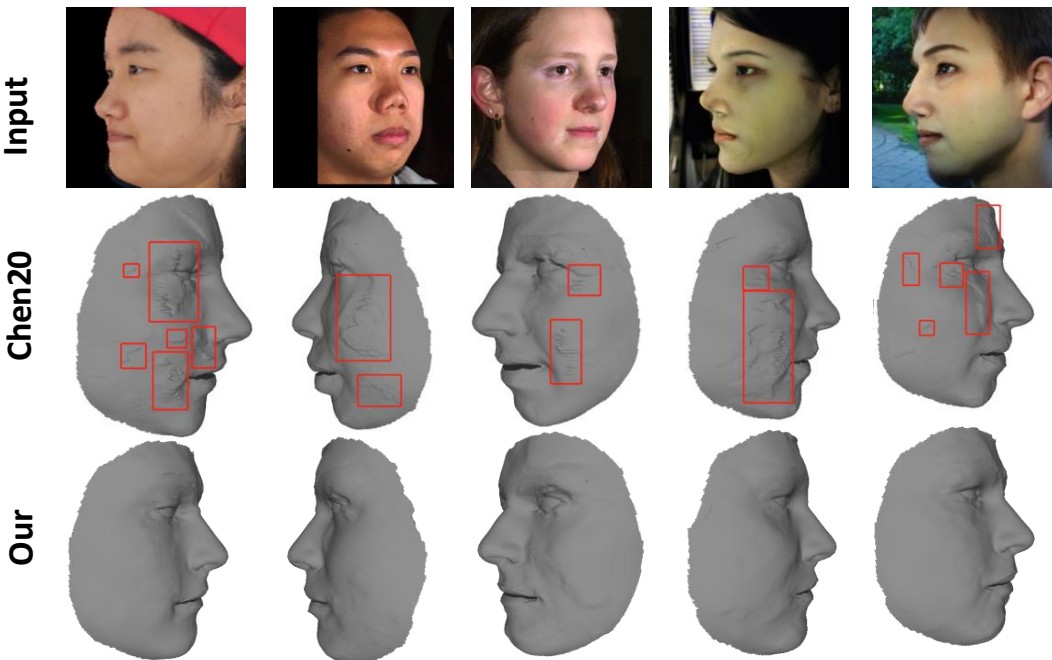

Figure 9: Visible generalization performance results.

## F.2 EVALUATION OF GENERALIZATION PERFORMANCE

We show the visible generalization performance results in figure 9. It could be seen that there are a large number of unreliable face details in the five results of the Chen20 Chen et al. (2020), and our corresponding results are more credible. This proves that our method has great generalization ability.

