# OpenReview forum: "Detailed 3D Face Reconstruction in Full Pose Range"
_ICLR.cc/2024/Conference — ICLR 2024 Conference Withdrawn Submission_

### Official Review · Reviewer_AJb8 · 2023-10-29

**Soundness:** 2 fair
**Presentation:** 3 good
**Contribution:** 2 fair
**Rating:** 3
**Confidence:** 3

**Summary:**

This paper presents the problem of detailed 3D face reconstruction in full pose range from a single-face image. It highlights the limitations of existing methods in accurately reconstructing shapes and believable details for large pose images. To address these challenges, the authors propose a self-augment mechanism and a self-supervised detail reconstruction method for large-pose images. The self-augment mechanism generates a set of large pose data for re-learning in the first stage, while in the second stage, a Recursive Image-to-image Translation Network is used to estimate the displacement of depth maps and the original details of the self-occluded areas of large pose images. The proposed method improves the reconstruction capability of coarse 3D face and enables detailed 3D face reconstruction in full pose range. Experimental results show that the method achieves comparable performance to state-of-the-art methods.

**Strengths:**

- The task of accurately reconstruct 3D shapes and believable details for large pose face images has some interesting for the community of single-face based 3D face reconstruction.
- The proposed idea of self-augment mechanism to generate large pose data for re-learning of 3DMM based coarse 3D face reconstruction is interesting.
- The proposed idea of using face symmetry prior to solve the self-occluded areas in large pose images has some novelty to me. In fact, this idea has be used in the task of large pose 3D face recognition, see the following reference [1]. And using this prior to solve the large pose problem related to face is also a common idea.
[1] G. Passalis, P. Perakis, T. Theoharis and I. A. Kakadiaris, "Using Facial Symmetry to Handle Pose Variations in Real-World 3D Face Recognition," in IEEE Transactions on Pattern Analysis and Machine Intelligence, vol. 33, no. 10, pp. 1938-1951, Oct. 2011, doi: 10.1109/TPAMI.2011.49.
- Experimental results demonstrate comparable performance to state-of-the-art methods, and better results for large pose images.

**Weaknesses:**

-The whole pipeline for two-stage based 3D face reconstruction has been widely used, including the coarse 3D face reconstruction using the 3DMM model, the image-to-image translation model for displacement depth map estimation, as well as the loss functions. Thus, the novelty of the paper is limited to me.

**Questions:**

-How about the computational cost (including the training and testing) of the proposed method compared to the other SOTA methods?
-How about the performance of this kind of two-staged based 3D face reconstruction method compared to the other methods based on generative models, such as the diffusion model based method?

---

### Official Review · Reviewer_upxb · 2023-10-30

**Soundness:** 2 fair
**Presentation:** 3 good
**Contribution:** 2 fair
**Rating:** 3
**Confidence:** 4

**Summary:**

This paper introduces a monocular detailed 3D face reconstruction method that primarily addresses the problem of effectively reconstructing the 3D details of the invisible areas in cases of large poses or facial occlusion.

**Strengths:**

The paper has a commendable motivation. Most existing monocular detailed 3D face reconstruction methods often overlook the reconstruction of 3D details of the invisible areas in cases of large poses or facial occlusion.

**Weaknesses:**

1. The paper lacks important comparison, excluding recent high-quality methods like HRN[1], FaceVerse[2], SADRNet[3], etc. Besides, the compared results are reproduced, not genenerated by the released codes of the original paper.
2. Some of the experimental settings in the paper appear to be somewhat unrealistic. For instance, the use of only 1000 images for training data deviates from the typical research experimental setups in this field. Given that 3D face reconstruction has practical applications in computer vision, this dataset size may be inadequate for real-world requirements. For example, DECA and Chen utilize significantly larger datasets, which raises questions about the data size used in this paper. For example, HRN[1] uses more than 400K images in total, DECA uses more than 2,000K images and Chen20 [4] uses more than 160K  images in their original settings.

3. The entire pipeline in this paper is unique in face applications; however, it lacks a discussion about its relationship with the representation learning techniques commonly used in computer vision. This omission might not be appropriate for an ICLR submission.


[1] Lei B, Ren J, Feng M, et al. A Hierarchical Representation Network for Accurate and Detailed Face Reconstruction from In-The-Wild Images[C]//Proceedings of the IEEE/CVF Conference on Computer Vision and Pattern Recognition. 2023: 394-403.
[2] Wang L, Chen Z, Yu T, et al. Faceverse: a fine-grained and detail-controllable 3d face morphable model from a hybrid dataset[C]//Proceedings of the IEEE/CVF conference on computer vision and pattern recognition. 2022: 20333-20342.
[3] Ruan Z, Zou C, Wu L, et al. Sadrnet: Self-aligned dual face regression networks for robust 3d dense face alignment and reconstruction[J]. IEEE Transactions on Image Processing, 2021, 30: 5793-5806.
[4] Chen Y, Wu F, Wang Z, et al. Self-supervised learning of detailed 3d face reconstruction[J]. IEEE Transactions on Image Processing, 2020, 29: 8696-8705.

**Questions:**

The authors should improve the paper's readability, for example, by adding comparisons of rendered facial reconstruction results, reducing complex sentence structures, and incorporating links for tables and figures.

The way the authors present their results also appears unusual. It seems that rendering the reconstruction results on the input image would be more advantageous for visualization.

---

### Official Review · Reviewer_7m2U · 2023-11-01

**Soundness:** 3 good
**Presentation:** 3 good
**Contribution:** 2 fair
**Rating:** 5
**Confidence:** 5

**Summary:**

This paper proposes a pipeline for recovering geometric details of 3D faces from images, especially those with large head poses. To achieve this goal, the author use the self-augment mechanism, by rotating and generating rendered face images with various large poses and re-use them as training data. A Recursive Image-to-image Translation Network is constructed for estimating details for the large-pose images, based on the face details reflected in the original image.

**Strengths:**

1. For generating reasonable geometric details, the proposed method does not require additional training images or 3D data.
2. Qualitative results show that compared to Chen20, the proposed method can generate subtle facial details such as wrinkles.
3. Clearly written, method understandable.

**Weaknesses:**

1. The discussion of related works and compared methods are insufficient. More state-of-the-art works should be discussed for detailed 3D face reconstruction. (For example, the NeRF model has been used to recovering 3D facial details.) Some unmentioned related works include  [1][2][3][4][5], etc.

2. Besides, the self-augment mechanism is not novel, generating augmented head pose images from 3DMM and re-use them for training has been considered and used in related tasks such as [6] and [7].

3. For facial detail reconstruction, the FaceScaoe dataset provide very high-resolution 3D meshes which should be considered for quantitative evaluation (only qualitative evaluation is provided in this paper) and we suggest the author move this evaluation in the main paper.

4. I’m also questioning about the self-augmented session, why using synthesized head pose images purely from 3DMM can improve the coarse reconstruction accuracy, as people can always generate arbitrary number of rendered large-pose images from 3DMM by randomizing the identity parameter. Also, I’m wondering why not using image UV map as the texture for the augmented coarse render image.

5. More recent works such as [1] and [2] should be compared for coarse or detailed reconstruction.


[1] Ziqian Bai, Zhaopeng Cui, Xiaoming Liu, and Ping Tan. Riggable 3d face reconstruction via in-network optimization. In Proceedings of the IEEE/CVF Conference on Computer Vision and Pattern Recognition, pages 6216–6225, 2021

[2]Daněček, R., Black, M.J. and Bolkart, T., 2022. EMOCA: Emotion driven monocular face capture and animation. In Proceedings of the IEEE/CVF Conference on Computer Vision and Pattern Recognition (pp. 20311-20322).

[3] Ramon, E., Triginer, G., Escur, J., Pumarola, A., Garcia, J., Giro-i-Nieto, X. and Moreno-Noguer, F., 2021. H3d-net: Few-shot high-fidelity 3d head reconstruction. In Proceedings of the IEEE/CVF International Conference on Computer Vision (pp. 5620-5629).

[4]Lei, B., Ren, J., Feng, M., Cui, M. and Xie, X., 2023. A Hierarchical Representation Network for Accurate and Detailed Face Reconstruction from In-The-Wild Images. In Proceedings of the IEEE/CVF Conference on Computer Vision and Pattern Recognition (pp. 394-403).

[5]Chatziagapi, A. and Samaras, D., 2023. AVFace: Towards Detailed Audio-Visual 4D Face Reconstruction. In Proceedings of the IEEE/CVF Conference on Computer Vision and Pattern Recognition (pp. 16878-16889).

[6] Wu, F., Bao, L., Chen, Y., Ling, Y., Song, Y., Li, S., Ngan, K.N. and Liu, W., 2019. Mvf-net: Multi-view 3d face morphable model regression. In Proceedings of the IEEE/CVF conference on computer vision and pattern recognition (pp. 959-968).
[7] Qin, J., Shimoyama, T. and Sugano, Y., 2022. Learning-by-novel-view-synthesis for full-face appearance-based 3D gaze estimation. In Proceedings of the IEEE/CVF Conference on Computer Vision and Pattern Recognition (pp. 4981-4991).

**Questions:**

Questions are listed in item 4 in *Weakness.

---

### Official Review · Reviewer_Zc9u · 2023-11-01

**Soundness:** 2 fair
**Presentation:** 2 fair
**Contribution:** 2 fair
**Rating:** 3
**Confidence:** 5

**Summary:**

The paper introduces an approach to tackle the problem of detailed 3D face reconstruction, especially for large pose images. It proposes a two-stage method. The first stage involves a self-augment mechanism, generating a more diverse dataset by rotating reconstructed 3D faces to simulate different poses. The second stage addresses detail reconstruction by padding self-occluded areas based on facial symmetry and employing a Recursive Image-to-image Translation Network. The proposed solution claims to alleviate constraints on coarse 3D face reconstruction training and recover more accurate face details across a full range of poses.

**Strengths:**

The paper proposed to tackle the challenging problem of detailed 3D face reconstruction in a full pose range.

**Weaknesses:**

1. Lack of Novel Technical Contributions: The paper lacks significant novel technical contributions as both "self-augment mechanism" and “self-supervised detail reconstruction” appear to be more of an engineering solution rather than a novel technique. The method relies on existing techniques and does not advance the state of the art in terms of algorithmic innovations.

2. The MICC dataset's precision level is not particularly high for the task of detailed 3D face reconstruction. Leveraging the MICC dataset for such evaluations is not convincing to me. A more appropriate choice would have been the FaceScape dataset, known for its higher precision and suitability for detailed 3D face reconstructions.

3. Visual Performance Concerns: Upon inspection of Fig. 6 (main paper) and Fig. 1 (supplementary material), the visual performance of the proposed method appears to be inferior when compared to the baseline "Self-supervised Learning of Detailed 3D Face Reconstruction (TIP2020)". This discrepancy between quantitative results and visual assessments raises concerns about the method's practical effectiveness.

4. The paper lacks consistent clarity and concise writing throughout. The language and sentence structures need improvement to ensure that the content is easily accessible to a wider audience. For example, "3D deformable model (3DMM)" should be "3D Morphable Model (3DMM)"

**Questions:**

1. What is the computational cost associated with implementing the method in terms of training time, data, and hardware requirements?

2. How do the proposed reconstruction methods perform on datasets with varying degrees of pose variations and with more diverse facial characteristics, such as expression and resolutio?